# Spinal Locomotion in Cats Following Spinal Cord Injury: A Prospective Study

**DOI:** 10.3390/ani11071994

**Published:** 2021-07-03

**Authors:** Ângela Martins, Cátia Marina Silva, Débora Gouveia, Ana Cardoso, Tiago Coelho, Óscar Gamboa, Eduardo Marcelino, António Ferreira

**Affiliations:** 1Faculty of Veterinary Medicine, Lusófona University, Campo Grande, 1300-477 Lisboa, Portugal; catiamsilva@outlook.com (C.M.S.); eduardo.marcelino@ulusofona.pt (E.M.); 2Animal Rehabilitation Center, Arrábida Veterinary Hospital, Azeitão, 2925-583 Setúbal, Portugal; dgouveia6@icloud.com (D.G.); anacardosocatarina@gmail.com (A.C.); tiagoccoelho@netcabo.pt (T.C.); 3CIISA—Centro Interdisciplinar-Investigação em Saúde Animal, Faculdade de Medicina Veterinária, Av. Universidade Técnica de Lisboa, 1300-477 Lisboa, Portugal; aferreira@fmv.ulisboa.pt; 4Superior School of Health, Protection and Animal Welfare, Polytechnic Institute of Lusophony, Campo Grande, 1300-477 Lisboa, Portugal; 5Faculty of Veterinary Medicine, University of Lisbon, 1300-477 Lisboa, Portugal; ogamboa@fmv.ulisboa.pt

**Keywords:** spinal cord injury, treadmill training, central pattern generator, spinal locomotion, reflexes, cats, neurorehabilitation

## Abstract

**Simple Summary:**

Functional neurorehabilitation promotes neural reorganization by stimulating subjects without deep pain perception, leading to a faster recovery when compared to spontaneous recovery, and achieving fewer compensatory errors, or even deviations to neuropathic or adaptive pain pathways, such as spasticity. The present study demonstrates the importance of intensive and repetition-based functional neurorehabilitation, which is essential for subjects classified as grade 0 according to the modified Frankel scale.

**Abstract:**

This article aimed to evaluate the safety and efficacy of intensive neurorehabilitation in paraplegic cats, with no deep pain perception (grade 0 on the modified Frankel scale), with more than three months of injury. Nine cats, admitted to the Arrábida Veterinary Hospital/Arrábida Animal Rehabilitation Center (CRAA), were subjected to a 12-week intensive functional neurorehabilitation protocol, based on ground and underwater treadmill locomotor training, electrostimulation, and kinesiotherapy exercises, aiming to obtain a faster recovery to ambulation and a modulated locomotor pattern of flexion/extension. Of the nine cats that were admitted in this study, 56% (*n* = 5) recovered from ambulation, 44% of which (4/9) did so through functional spinal locomotion by reflexes, while one achieved this through the recovery of deep pain perception. These results suggest that intensive neurorehabilitation can play an important role in ambulation recovery, allowing for a better quality of life and well-being, which may lead to a reduction in the number of euthanasia procedures performed on paraplegic animals.

## 1. Introduction

Locomotion is based on the synchronization between the flexion and extension of the limbs and reflex circuits of the spinal cord. However, the spinal cord alone is not sufficient to enable walking, and the input of the brain centers is also necessary [1]. The alternation between the extension and flexion of the pelvic limbs involves the central pattern generators (CPGs). The CPGs are spinal locomotor circuits with pacemaker proprieties, located in the thoracolumbar spinal cord, which generate bilateral rhythmic and repetitive contractions and relaxations of the flexors and extensor muscles in the absence of descending motor tracts and supraspinal inputs [1,2]. The supraspinal control of the CPGs originates in the nuclei, located in the midbrain (“mesencephalic locomotor region”). These nuclei initiate locomotion by activating the reticulospinal neurons in the brain stem. In cats, there are the following two major descending motor tracts: the medial longitudinal fascicle, with cells originating in the medial pontomedullary reticular formation, and the lateral vestibulospinal tract [3].

Locomotor training after spinal cord injury is based on the principle that sensory stimuli reactivate and reorganize the spinal locomotor circuit [4]. Over the last decade, the expression of hind limb locomotion after Spinal Cord Injury (SCI) at T13 in kittens [5,6] and adult cats has been investigated [7,8]. After 2–3 weeks of land treadmill training, both were able to walk and recover full-weight-bearing. To modulate the flexor–extensor pattern, some researchers suggest the association of pharmacologic management [9,10]. Thus, the adult cat spinal cord, in vivo, was proposed as a classical model in studies of mammalian hindlimb CPGs [11,12,13], allowing the translation of these findings to a clinical setting.

In this study, intensive neurorehabilitation is used to promote neural locomotion control. This control is based on spinal cord autonomy, the influence of sensory stimuli on the spinal cord autonomy, neuromodulation, the learning capacity of the spinal cord locomotor circuit, and, finally, the importance of the descending pathways in the control of locomotion [14]. This paper aimed to evaluate the safety and efficacy of intensive neurorehabilitation in grade 0 paraplegic cats, with more than three months of injury, in a clinical environment.

## 2. Materials and Methods

This prospective clinical study was conducted in HVA/CRAA, from 15 September 2014 to 15 March 2021, according to the Faculty of Veterinary Medicine, ULHT ethics committee, after the owner’s consent.

### 2.1. Characterization of the Population

For the present study, only cats with a severe contusion of traumatic origin, hit by a car or injured from a fall, were included. The diagnosis was obtained by radiology and myelotomography. At 24/48 h after SCI, imaging exams showed edema (a presence of hypoattenuating fluid/tissue adjacent to the lesion site) and an absence of vertebral spine damage (fracture and/or luxation). The lesions were localized between T3 and L3 and had to be more than 3 months old. In addition, cats had to exhibit grade 0 in the modified Frankel scale (MFS), i.e., be paraplegic with an absence of deep pain perception (DPP) and a lack of superficial sensitivity (SS) at the level of the tail (base and tip) and perineum. The upper motor neuron (UMN) injury led to a neurogenic bladder.

Cats with compressive myelopathies, intervertebral disc disease (IVDD), mainly spinal cord extrusion, neoplastic disease, and others were excluded from the study. Cats presenting severe contusions with neurolocalization beyond T3–L3 as well as cats with contusions at T3–L3, but with a degree higher than 0, according to MFS, were also excluded.

### 2.2. Plan of the Study

Upon admission to the study, the neurorehabilitation consultation was performed, and the subjects were redirected to CRAA. The intensive neurorehabilitation protocol (INRP) was initiated 24 h after admission, allowing for accommodation to the new environment. The INRP was performed 6 days a week, from 10 a.m. to 6 p.m. In addition to the re-evaluations of the INRP in accordance to the neurological development, cats were evaluated over a period of 2 to 3 months, always by the same person, in the same environment, and at the same time, with a video recording of steps taken on alternate floor types, using a Canon EOS Rebel T6 1300D camera. The reevaluation was only recorded at specific time points, as described later. The cats that reached ambulation before 3 months (time limit for INRP) were discharged from the hospital and redirected to the neurorehabilitation follow-up appointment.

### 2.3. Neurorehabilitation Consultation at Study Admission

Cats were classified as grade 0 in the MFS and as paraplegic. Peripheral spinal reflexes were evaluated through percussion with Taylor’s hammer. In the evaluation of peripheral spinal reflexes for the pelvic limb, the patellar reflex (PR), the tibial cranial reflex (TCR), the crossed extensor reflex (CER), and the flexor reflex (FR) were assessed. With cats in sternal recumbency, the cutaneous trunci reflex (CTR) was evaluated. Contracture or shortening of the femoral quadriceps muscle, associated with hypotonicity of the flexor muscle group, was observed in all subjects. The muscle tone of the rectus abdominis (RA) was also evaluated. Finally, sensitivity to superficial pain was assessed, i.e., the dermatomes of the pelvic limb, followed by an evaluation of deep pain perception in the medial and lateral digit of each limb. Twelve centimeter straight-ended Halstead mosquito forceps were used for this procedure.

### 2.4. Intensive Neurorehabilitation Protocol

The INRP consisted mainly of locomotor training and electrostimulation. Regarding locomotor training, there were the following two modalities: bipedal locomotor training (BLT) and quadrupedal locomotor training (QLT) (Table 1).

In BLT, the subject was placed on the land treadmill with the thoracic limbs on a smooth surface platform, and gait movement was assisted only in the pelvic limbs. The therapist sat behind the cat and performed cycling movements at a constant pace. Initially, the BLT was executed 3–6 times a day for 2–5 min, increasing progressively to 3 times a day, for about 20 min, 6 days a week. Regarding the velocity of the treadmill, in the initial stage, we started at 0.8 km/h (0.22 m/s) and increased to 1.2 km/h (0.33 m/s), without a slope.

In the first stage, QLT (Figure 1) was performed 4–8 times a day for 2–5 min, aiming at sessions longer than 30 min. In the final stage, the training advanced to 3 times a day, 6 days a week. The velocity of the treadmill was between 1 km/h (0.27 m/s) and 1.8 km/h (0.5 m/s), with a 10–25% slope.

In both modalities, the therapist stimulated the perineal region and base of the tail to induce stepping and progressively supported the weight by holding the tail. In the initial stage, the therapist performed cycling movements in alternation with the perineal and base of the tail stimulation.

In the underwater treadmill locomotor training (UWT), cats were fitted on a harness in such a way that a guidance rope system was provided by the rehabilitation technician outside the underwater treadmill. This type of training was conducted once a day, 5 days a week, from 5 min up to 40 min, according to subject compliance, with a 10% slope. The temperature of the water was 24–26 °C, and after UWT, cats were dried with an absorbent towel in a calm environment.

To complement the locomotor training, passive, passive-assisted, and active kinesiotherapy exercises were administered, such as postural standing stimulation (2–5 min, 4–6 times a day), gradually introducing difficulties with postural standing on the trampoline or on balance boards (3–5 times a day, 5 days a week) (Figure 2), and on cavalettis rails, for 2 min, 2–4 times a day, 5 days a week.

The locomotor training was associated with electrical stimulation. Segmental electrical stimulation was prescribed according to the sciatic nerve path, with one electrode in the lumbosacral anatomical region (L7—S1/S2 )and the other in the ventral medial region of the thigh, close to the motor point of the hamstring muscles (biceps femoris, semitendinosus, and semimembranosus) (Figure 3). The parameters of the pulsed biphasic electric current were 40–60 Hz, up to 24 mA, with a working cycle from 1 shot to 4 at rest (1:4) and a ramp 4 s upward, 8 s at the plateau, and 2 s downward. This was executed twice a day, for 10 min, 5 days a week. Milliamperage was applied with a constant observation of the muscle so that the limit was a maximum involuntary contraction, but still comfortable for the subject.

### 2.5. Monitoring of the Study Population

All cats were evaluated at the time of inclusion in the study (M0). The neurological evolution up to the ambulatory state was evaluated after 15 days of hospitalization (M1) and then, reevaluated after 4 weeks (M2), 8 weeks (M3), and 12 weeks (M4) of hospitalization. Follow-up appointments were made 15 days (R1), 1 month (R2), 3 months (R3), 6 months (R4), 1 year (R5), 2 years (R6), and 3 years (R7) after medical discharge. The temporal space between neurorehabilitation appointments is schematically shown in Figure 4.

At every appointment (from M0 to M4 and from R1 to R7), cats were evaluated according to the following: active postural standing capacity, the presence of a flexor reflex, a crossed extensor reflex, CTR, rectus abdominis muscle tone, superficial sensitivity in the perineum and tail (base and tip), evaluations of deep pain perception in the medial and lateral digits, and gait stimulation on different floors.

Ambulation is related to motor capacity, whereby one can perform more than 10 steps independently of the floor and perform curves with a greater support base, demonstrating slight squat secondary to muscle weakening, without ever falling, to perform, for example, micturition, automatic or voluntary urination.

## 3. Results

The database and statistical analyses were performed with Excel 2013 ^®^ (Microsoft, Redmond, WA, USA).

In this study (*n* = 9), the cats had a mean age of 3.9 years, 67% (*n* = 6) were female and 33% (*n* = 3) were male, with a mean body weight of 4.4 kg.

Four cats (44%) presented the flexor reflex bilaterally, four (44%) presented a diminished flexor reflex, and one (11%) presented an exaggerated flexor reflex. After the INRP, 89% (*n* = 8) of the cats presented a flexor reflex, and only one (11%) presented a clonic flexor reflex, as seen in Table 2.

Regarding the CTR, at admission, 56% (*n* = 5) had a CTR at the lesion site (note that a lesion is approximately two vertebral bodies cranial to the cutoff point), 33% (*n* = 3) exhibited an absence of a CTR at the lesion site, and one showed a CTR in the lumbosacral region. Following the INRP, 56% (*n* = 5) demonstrated a recovery, with the CTR in the lumbosacral region, 33% (*n* = 3) presented a CTR at the lesion site, and one showed no signs of neurological improvement, with the CTR remaining absent at the site of the lesion.

Regarding the neurological grade (Table 3), all of the cats (*n* = 9) were paraplegic, without DPP, at admission; whereas, after undergoing INRP, 44% (*n* = 4) of the cats presented functional spinal locomotion by reflexes, i.e., ambulation, 22% (*n* = 2) had non-functional spinal locomotion by reflexes, 22% remained paraplegic without DPP, and 11% (*n* = 1) had recovered DPP and stayed paraparesic ambulatory with proprioceptive ataxia.

In the study, non-functional spinal locomotion was considered when the cats exhibited a flexor-extensor pattern, moving on the floor, or with weight support by holding the tail, without being able to stand up.

In 44% (*n* = 4) of the cats, the remodeling regarding superficial sensitivity was evident, because three cats had superficial pain at the perineum, although two of them also presented this at the tip and base of the tail. Only one cat regained DPP, although when comparing the deep pain of the thoracic limbs, a slower reaction was found. The results of the present study are summarized in Figure 5.

## 4. Discussion

This study used nine cats who were diagnosed with a severe contusion of traumatic origin, in a chronic state, i.e., more than three months after injury. Since spontaneous neurological recovery after injury reaches a plateau, usually within the first few months, the neurological evolution observed is expected to be the consequence of the INRP, rather than the continuation of spontaneous recovery [15,16]. Furthermore, the fact that stimulation through INRP was extended for three months, and that the plateau that is indicative of motor recovery occurs between two and three months [7,17,18], also accounts for the INRP for the neurologic improvement of cats.

After spinal cord injury, both humans and cats can experience spontaneous recovery that is observed even without functional rehabilitation, and can be explained by the same cellular mechanisms that can occur in spontaneous remodeling [19,20,21]. This makes it difficult to separate, concerning the clinical signs that are indicative of neurological recovery over time, the evolution that is mainly due to the action of rehabilitation training from that of spontaneous recovery [22].

Early and important studies in neuroscience were performed in acute and chronic adult spinal cats and were based on a complete section of the spinal cord in a laboratory setting, which resulted in the paralysis of the pelvic limbs (grade 0 in MFS) [4,7,8,23,24]. The breakthrough of intensive rehabilitation in SCI was based on the studies mentioned above [22]. Furthermore, training procedures that showed small stepping movements on the land treadmill led to a marked improvement in the locomotor performance [25], although it appears that the type of locomotor training was the most important factor in the recovery of locomotion [26].

The main difference between those studies, when compared to ours, is the velocity applied on the treadmill. In our study, locomotor training was performed, in some cases, bipedal training that progressed to quadrupedal training, which allowed the spinal cats to adapt to the treadmill and made it possible to reach higher treadmill speeds. Lovely et al., 1990 [26] achieved stepping on the land treadmill with velocities of 0.05–0.15 m/s^−1^, which was a slow speed when compared to our study (0.22 m/s increasing to 0.5 m/s). The maximum land treadmill speed at which spinal locomotion could be obtained was reported from 0.5 to 0.6 m/s, only after three weeks of training [27]. Our study accords with these training procedures.

Lovely et al., 1990 [26] also performed daily locomotor training, similar to our protocol, but with an early training approach in the acute phase, justifying their results. The same study showed that a period of four to six months was needed to achieve ambulation, while Barbeau and Rossignol (1987) proved that acute spinalizated adult cats (T13) were capable of demonstrating a gait with the weight support of the hindlimbs without showing knuckling, from three weeks up to one year [7]. On the other hand, Edgerton et al., 1991 [28] performed 30 min of daily training for six months, reaching a performance plateau at tow to four months. Regarding time, this training was similar to the one implemented in our protocol, although 33% (*n* = 3) reached a plateau performance in one month.

Barbeau and Rossignol, 1990 [29] introduced pharmacological management in cats between one and three months after spinalization. The cats reached a treadmill velocity of 1.0 m/s before and after an injection of a serotonergic monoamine drug (5-HT). They concluded that the injections of 5-HT increased the cycle duration by about 80%, and that noradrenergic, dopaminergic, and GABAergic drugs could improve the initiation of locomotion (in an early stage), and modulate the well-established locomotor pattern (in a chronic stage). In addition, studies were performed on the association between pharmacologic management and FES [30]. In this regard, some results showed that electrical stimulation could initiate or modify locomotor activity in a similar way to pharmacological neural modulation [25]. These findings show that the use of electrostimulation can be a neurorehabilitation modality, used to enhance the restoration of full-weight-bearing locomotor function [31].

Smith et al., 1982 [32] demonstrated that spinal cats could exhibit excellent weight support during locomotion, applying treadmill speeds of ≤0.8 m/s, which suggested velocity as a major factor in the recovery of ambulation. In addition, the results have shown that, in the first 7–10 days post-section, cats only executed small hindlimb movements, with little to no weight support, which could be enhanced with pharmacological treatment, stimulating the central plasticity that could be reflected by the evolution of the locomotor pattern over time [10].

When compared to our study, on both modalities of the locomotor training (BLT and QLT), combined with FES, ambulation was obtained in 56% of cats (*n* = 5). Thus, only 10% of the white matter of the spinal cord is required to initiate and maintain the locomotor pattern, in a voluntary way, on the land treadmill [33], which can probably be justified by the residual descending motor tracts post-injury [34]. In our study, one cat recovered their DPP, obtaining medical discharge after two weeks of INRP. As for the other three cats exhibiting SS, two had medical discharge one month after INRP and the other one did two months afterwards. It is suggested that severe injury contains sub-functional connections that are capable of transmitting a supraspinal influence on the neural circuits of CPG, below the injury [35,36]. Following the researches of Dimitrijevic et al. 1987, Militskova et al. refers to this type of lesion as “discomplete SCI”, which agrees with Gerasimenko et al., 2017 [37], who suggest that locomotor training can significantly improve sensorimotor and autonomic function after SCI [38].

In the neurophysiological field, it has been proposed that the use of an α2-noradrenergic agonist, such as clonidine, could initiate treadmill locomotion faster in cats with acute and chronic spinal injury [39], and modulate the spinal locomotor pattern [10], allowing fictive locomotion, which may be recovered spontaneously in several weeks. Though in chronic spinal cats, the activation of the receptors does not express the spontaneous spinal locomotion [40], excitatory amino acid receptor agonist NMDA, injected intrathecally, could result in a dramatic improvement in the locomotor pattern.

In some studies, fictive locomotion has been achieved. This fictive locomotion could be defined as a rhythmic pattern that occurs in the absence of any movement [2,41], and in chronic spinal cats, it can occur spontaneously, which indicates functional changes in the interneurons and CPGs [4,42].

Our strict INRP suggested that BLT could activate proprioceptive afferents (groups Ia, Ib, and II) [42,43] throughout stretch bicycle movements, performed by the therapist, in alternation with the stimulation of the perineal region [44] and the base of the tail [19], which regulated, in some part, the duration of various sub-phases of the step cycle (frequency and speed). Moreover, the hip joint influences the locomotor rhythmic generation and has a potent effect on the intrinsic neural circuits, because, when the hip is flexed, the spinal cat stops the stepping movement [45]. Furthermore, the QLT may activate the CPG that regulates the fore- and hindlimb locomotion, and this can be blocked when the propriospinal pathways that connect the cervical and lumbar enlargements are interrupted [46,47]. Thus, the QLT stimulates the coordinating propriospinal system, providing forelimb–hindlimb coordination [48]. In our study, the cat that showed recovery of DPP could perform QLT in the first instance.

The QLT presumably increases the synaptic activity by interneuron stimulation, which is responsible for locomotor patterns and intrinsic electrophysiological proprieties [49,50] that are vital to neural control [51]. Locomotor training can have an impact on spinal cord autonomy, influence sensory inputs, induce neuromodulation and learning abilities of the spinal network, promote depolarization of the descending motor pathways [51], and upregulate the neurotrophins, particularly the neurotrophic factor derived from the brain (BDNF) that plays an important role in central nervous system (CNS) neuroplasticity, contributing to the restoration of function [25]. The intense repetitive training may have driven neuroplasticity, which eventually results in the ability of voluntary movement, by reorganizing its structure, function, and connections [52,53,54,55,56].

The QLT promotes stimulation of the afferent pathways by receptors, located in the muscles (intrafusal fibers), joints (nerve endings), and skin. This type of stimulation allows a dynamic interaction with the CPG circuit [3,14,57], and cutaneous neural stimuli in the digit region that stimulates the expression of reflex locomotion, allowing the observation of neural reorganization after four weeks of locomotor training on the treadmill [7,19,58]. During the exercises, several interconnected components are activated, which are necessary to obtain correct locomotion [4].

The vestibulospinal and reticulospinal tracts are responsible for posture, meaning that they have a greater positive effect on the extensor muscles [59], so the introduction of the slope during QLT will stimulate these muscles and, therefore, the neuroplasticity of the tracts mentioned above. Thus, a 10° slope is suggested by Maier et al., 2009 [60], and 25° is suggested by Tillakaratne et al., 2014 [61].

Locomotor training that allows neural plasticity should be performed for 30–60 min [14,20], which is in agreement with the INRP presented in this study. The INRP is associated with kinesiotherapy exercises that play a role in synaptic neuroplasticity, in the sense of neuroremodeling [62,63]. To achieve a balance between excitation and inhibition, in the passage of an obstacle, there is a stimulation of receptors located in the distal dorsal region of the limb that will allow excitation of the flexor muscle group of the ipsilateral limb, when performing the protraction phase, and that, with the same stimulus, can excite the extensor muscles that are necessary to obtain the postural phase of the step cycle [64]. An example of the above is active-assisted and/or active kinesiotherapy exercises, which are included in locomotor training [65]. Included in the INRP are the cavalettis rails for stimulating the passage of obstacles.

The cat that had recovered DPP showed fast progression when performing these exercises, supporting the notion that DPP is considered a favorable prognostic factor [34,66,67,68]. Therefore, the ability to execute early locomotor training, reaching the mentioned performance guidelines, may suggest a higher probability of recovery ambulation in a shorter amount of time.

The limitations of our study are the small population sample and the fact that the cats had greater difficulties adapting to the INRP, requiring a longer period.

## 5. Conclusions

The present prospective study demonstrates the importance of intensive and repetition-based functional neurorehabilitation in spinal cord injuries, since the INRP allowed 56% of chronic paraplegic cats with no DPP to recover ambulation with functional spinal locomotion reflexes.

The protocols of intensive functional neurorehabilitation are safe and can allow a significant improvement in the neurological state of spinal cats.

Thus, in cats with grade 0 in the MFS, it is suggested that early locomotor training for at least 30–60 min, attaining speeds of 0.5–0.6 m/s, associated with FES, is needed, allowing for the stimulation of the descending motor tracts and the afferent inputs, inducing neuromodulation, and promoting the learning ability of the spinal network. Given its limitations, this study should be continued in a clinical setting.

## Figures and Tables

**Figure 1 animals-11-01994-f001:**
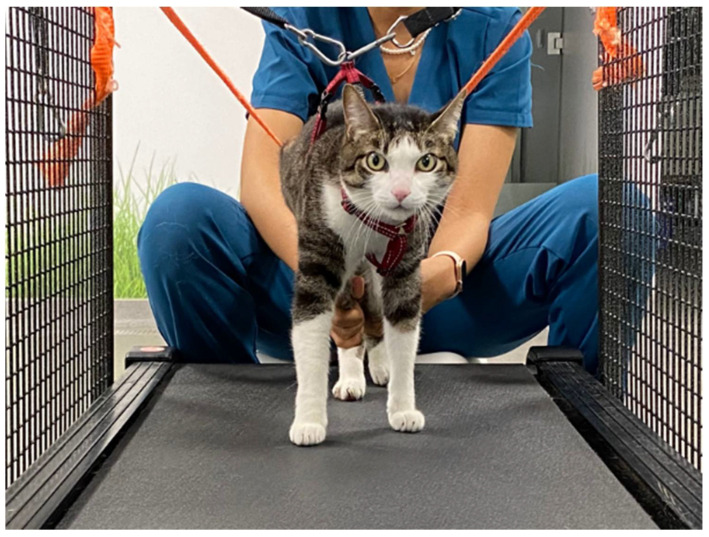
Quadrupedal locomotor training (QLT) performed on a land treadmill.

**Figure 2 animals-11-01994-f002:**
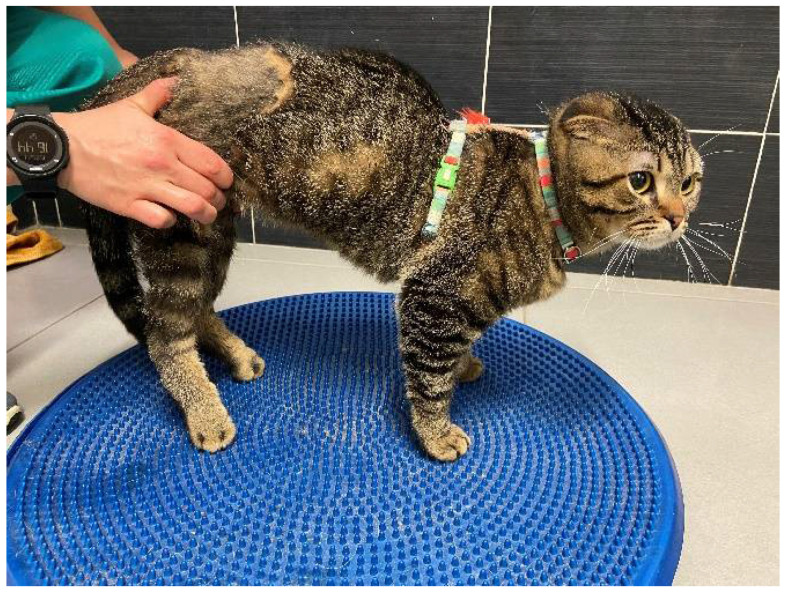
Kinesiotherapy exercises—postural standing stimulation on a balance board.

**Figure 3 animals-11-01994-f003:**
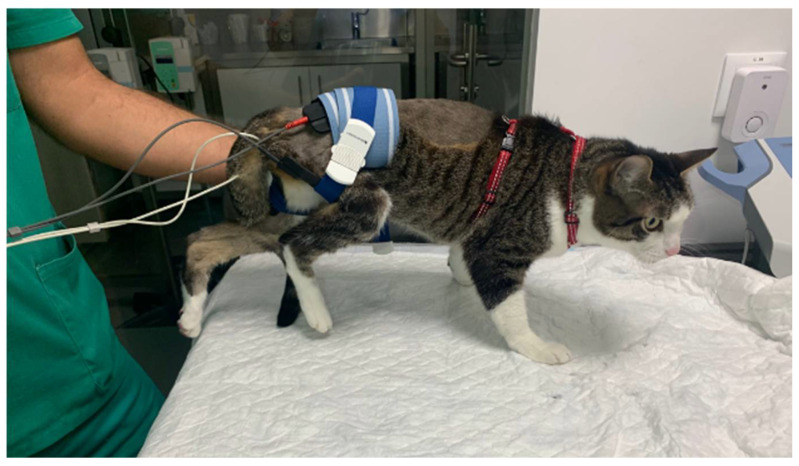
Functional electrical segmental stimulation of the sciatic nerve.

**Figure 4 animals-11-01994-f004:**
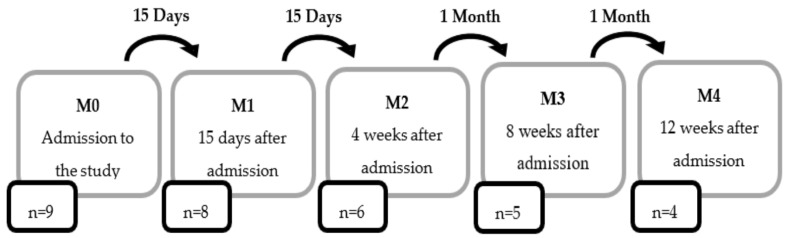
The schematization of the time interval between neurorehabilitation appointments.

**Figure 5 animals-11-01994-f005:**
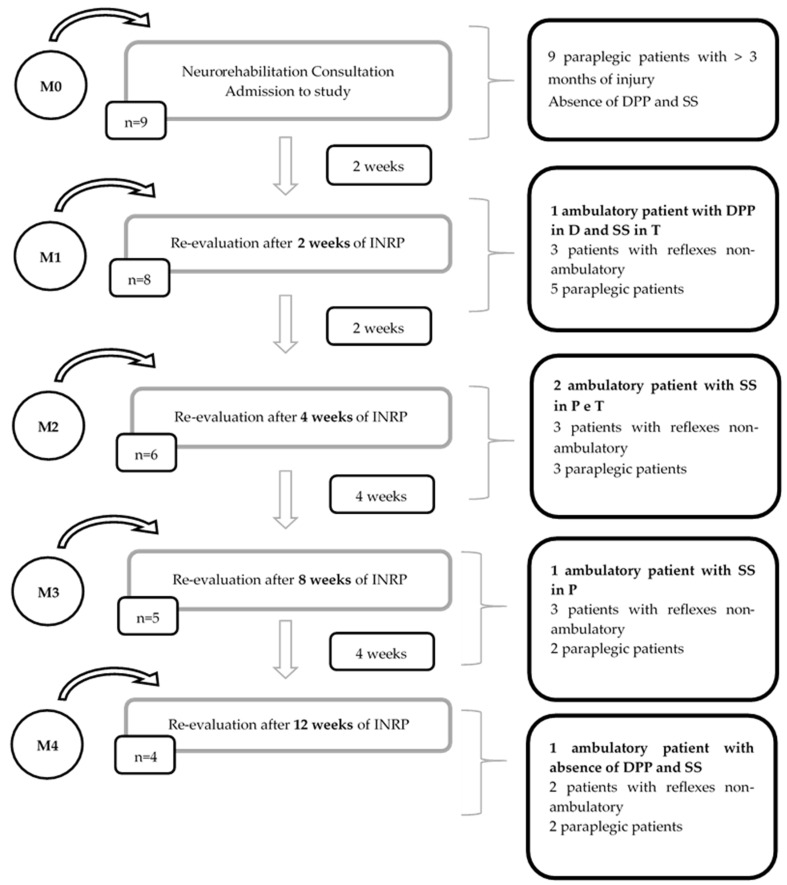
Patient progression algorithm in the study. DPP: deep pain perception; SS: superficial sensitivity; P: perineum; T: tail; D: digits.

**Table 1 animals-11-01994-t001:** Locomotor training performed by the cats.

	Locomotor Training	
Animal	Treadmill	Underwater Treadmill
I	Bipedal first 15 days and then quadrupedal	Bipedal first 15 days and then quadrupedal
II	Bipedal	Not tolerated
III	Bipedal first 15 days and then quadrupedal	Bipedal first 15 days and then quadrupedal
IV	Bipedal first 15 days and then quadrupedal	Bipedal first 7 days and then quadrupedal
V	Quadrupedal	Not tolerated
VI	Quadrupedal	Quadrupedal
VII	Bipedal first 15 days and then quadrupedal	Not tolerated
VIII	Bipedal first 15 days and then quadrupedal	Not tolerated
IX	Bipedal	Bipedal

**Table 2 animals-11-01994-t002:** Classification of peripheral spinal reflexes and muscle tone of the rectus abdominis, before and after the application of INRP (medical discharge).

Animal	FR		CTR		CER		RA	
	Before INRP	After INRP	Before INRP	After INRP	Before INRP	After INRP	Before INRP	After INRP
I	Present bilateral	Normal	Present at the site of lesion *	Recovery (since the lumbosacral region)	Present bilateral	Normal/Modeled with flexor reflex	Normal	Normal
II	Present bilateral exaggerated	Clonic	Present at the site of lesion *	Present at the site of lesion *	Present bilateral	Normal/Modeled with flexor reflex	Decreased	Normal
III	Present bilateral	Normal	Present at the site of lesion *	Recovery (since the lumbosacral region)	Present bilateral	Normal/Modeled with flexor reflex	Decreased	Normal
**IV**	Present bilateral decreased	Normal	Present since the lumbosacral region	Recovery (since the lumbosacral region)	Present bilateral decreased	Normal/Modeled with flexor reflex	Decreased	Normal
**V**	Present bilateral decreased	Normal	Present at the site of lesion *	Recovery (since the lumbosacral region)	Present bilateral	Normal/Modeled with flexor reflex	Decreased	Normal
**VI**	Present bilateral	Normal	Present at the site of lesion *	Recovery (since the lumbosacral region)	Present bilateral	Normal/Modeled with flexor reflex	Normal	Normal
**VII**	Present bilateral	Normal	Absent at the site of lesion	Present at the site of lesion *	Present bilateral decreased	Decreased	Decreased	Normal
**VIII**	Present bilateral decreased	Normal	Absent at the site of lesion	Present at the site of lesion *	Absent	Decreased	Decreased	Decreased
**IX**	Present bilateral decreased	Normal	Absent at the site of lesion	Absent at the site of lesion	Present bilateral decreased	Decreased	Decreased	Decreased

FR: flexor reflex; CTR: cutaneous trunci reflex; CER: crossed extensor reflex: RA: muscle tone of the rectus abdominis; * lesion is approximately two vertebral bodies cranial to the cutoff point.

**Table 3 animals-11-01994-t003:** Ambulation capacity at the end of the study.

	Neurological Grade before INRP Neurological Grade after INRP	
I	Paraplegic	Functional spinal locomotion by reflexes
II	Paraplegic	Non-functional spinal locomotion by reflexes
III	Paraplegic	Functional spinal locomotion by reflexes
IV	Paraplegic	Paraparesis ambulatory with proprioceptive ataxia
V	Paraplegic	Functional spinal locomotion by reflexes
VI	Paraplegic	Functional spinal locomotion by reflexes
VII	Paraplegic	Non-functional spinal locomotion by reflexes
VIII	Paraplegic	Paraplegic
IX	Paraplegic	Paraplegic

## Data Availability

The data presented in this study are available upon request from the corresponding author.

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
