# Peer review of "Spinal Locomotion in Cats Following Spinal Cord Injury: A Prospective Study"

_animals, 2021, doi:10.3390/ani11071994_

Round 1

Reviewer 1 Report

General comments:

Along the last decades it has been experimental established that after a complete spinal cord injury at T13 level in cats, hindlimb stepping can be reinstated by a wide variety of treatment methods. Several weeks of locomotor training (Barbeau and Rossignol, 1987; Lovely et al., 1990), pharmacological stimulation (Barbeau and Rossignol, 1990, 1994; Chau et al., 1998b; Giroux et al., 2003), or various combinations of these treatments (Barthe´lemy et al., 2006, Chau et al., 1998) can produce satisfactory recovery. Using intensive neurorehabilitation intensive protocols the authors provided promising results for paraplegic cats without deep pain sensation following spinal cord injury. In general the experimental design is innovative but very confusing. My main concern is about the discussion section. The authors must expand the discussion, comparing their results with the literature… please see above some of the most important authors. This manuscript should be more focused on treadmill training. It is not possible to have a clear evidence of the impact of the electrical stimulation in the functional recovery. In the discussion the authors should define fictive locomotion, a term used in the neurophysiology field.

Specific comments:

  1. Title: The authors should change the tittle of the manuscript. For example: “Spinal locomotion in cats following spinal cord injury. A prospective study”.
  2. Keywords: add “spinal cord injury”; “treadmill training”, “central pattern generator”.
  3. Introduction: please provide a definition of central pattern generator.
  4. Materials and Methods, 2.1.: What is a non-compressive myelopathy for authors? Is it a contusion type lesion without luxation/fracture? It would be interesting to see some CT/Myelograms.
  5. Materials and Methods, line 79: Is it “without urinary incontinence”?
  6. Materials and Methods, line 79: “2.2.” instead of “2.1.”
  7. Materials and Methods, line 88: please give more details about the camera used for gait analysis (model, frequency…).
  8. Materials and Methods, line 92: “2.3.” instead of “2.1.”
  9. Materials and Methods, lines 100-103: Why the authors assessed superficial pain in cats without dep pain perception?
  10. Material and Methods, line 106: They refer 2 modalities, but only described one, and the second one on another paragraph…
  11. Materials and Methods, lines 117-118: This sentence should be removed from the materials and methods section.
  12. Figure 4: The timing for evaluation, depicted in this figure, is different from line 87.
  13. Table 1: “CTR” instead “TCR”.
  14. Table 1: About CTR, the authors described “Present at the site of the injury”. This is not correct, as the level of spinal cord lesion is approximately 2 vertebral bodies cranial to the cutoff point.
  15. Table 1: About CTR, the authors described “Present above the injury”, so the CTR is normal.
  16. Results, line 182: I think it is Table 2…
  17. Results, lines 183-184: Please define non-functional spinal locomotion.
  18. Table 3: Looks a training protocol to be included in the materials and methods section.
  19. Results, lines 193-195: This sentence should be removed from the results section. They must explained these conclusions in the discussion.
  20. Discussion, lines 231-232: They must explained these conclusions in a scientific way.

Reviewer 2 Report

Please see file attached

Round 2

Reviewer 1 Report

General comments:

All the changes have been made by the authors accordingly to my revision.